# Foveal Intraretinal Fluid Localization Affects the Visual Prognosis of Branch Retinal Vein Occlusion

**DOI:** 10.3390/jcm11123540

**Published:** 2022-06-20

**Authors:** Hirofumi Sasajima, Masahiro Zako, Rio Maeda, Kenta Murotani, Hidetoshi Ishida, Yoshiki Ueta

**Affiliations:** 1Department of Ophthalmology, Shinseikai Toyama Hospital, Imizu 939-0243, Japan; r-maeda@shinseikai.or.jp (R.M.); ishi1214@kanazawa-med.ac.jp (H.I.); ueta@shinseikai.or.jp (Y.U.); 2Department of Ophthalmology, Asai Hospital, Seto 489-0866, Japan; mzako@aol.com; 3Biostatistics Center, Kurume University, Kurume 830-0011, Japan; kmurotani@med.kurume-u.ac.jp; 4Department of Ophthalmology, Kanazawa Medical University, Kahoku 920-0293, Japan

**Keywords:** branch retinal vein occlusion, visual prognostic factor, foveal intraretinal fluid, ellipsoid zone, macular edema, optical coherence tomography

## Abstract

We investigated whether baseline foveal intraretinal fluid (IRF) localization affects the visual prognosis of branch retinal vein occlusion (BRVO). Fifty eyes from 50 patients were included in this retrospective study. We classified the eyes with IRF involving and not involving the central foveola on the vertical optical coherence tomography (OCT) image at the initial visit into both-sides (*n* = 17) and one-side IRF (*n* = 33) groups, respectively. Multiple regression analyses demonstrated that not only the baseline logarithm of the minimum angle of resolution (logMAR) best-corrected visual acuity (BCVA) but also the IRF localization significantly correlated with the 12-month logMAR BCVA (*p* = 0.04 and *p* = 0.001, respectively), indicating that eyes with better baseline logMAR BCVA and one-side IRF have a significantly better visual prognosis in BRVO. The foveal ellipsoid zone band was significantly more disrupted (*p* < 0.001) in the both-sides IRF (47.1%) group than in the one-side IRF (3.0%) group. No eyes with decimal BCVA less than 0.5 were detected in the one-side IRF group at 12 months. Thus, baseline foveal IRF localization on vertical OCT images can be considered a novel biomarker for the visual prognosis of BRVO.

## 1. Introduction

Elucidation of potential visual prognostic factors is important since it could contribute to patient counseling and treatment strategies. Several potential visual prognostic factors for branch retinal vein occlusion (BRVO) have been reported. Patient age [1,2,3,4] and baseline logarithm of the minimum angle of resolution (logMAR) best-corrected visual acuity (BCVA) [1,2,3] are potential visual prognostic factors for BRVO.

High-resolution optical coherence tomography (OCT) has revealed several visual prognostic factors for BRVO, such as baseline central subfield thickness (CST) [3]. The correlation of CST with final visual acuity (VA) is somewhat controversial. Iida-Miwa et al. [3] reported that the baseline central foveal thickness is associated with the final VA. Meanwhile, other studies reported contradictory results [5,6,7,8].

OCT has helped identify other visual prognostic factors for BRVO, including integrity of the ellipsoid zone (EZ) [8,9] and external limiting membrane (ELM) [8,9] and disorganization of the retinal inner layers (DRIL) [10]. Sometimes, in acute-phase BRVO with severe retinal hemorrhage and macular edema (ME), evaluating the integrity of the EZ and ELM and the presence of DRIL on OCT images can be challenging. In comparison, determining whether intraretinal fluid (IRF) involves the central foveola through OCT can be easy in acute-phase BRVO.

We sometimes encounter eyes with IRF involving the central foveola, which have poor visual prognosis, in our clinical practice. These eyes with poor VA often show foveal ELM and EZ disruptions on OCT images after complete resolution of ME. Meanwhile, eyes with IRF not involving the central foveola usually do not display foveal ELM and EZ disruptions on OCT images after complete resolution of ME. Since IRF involving the central foveola could cause foveal photoreceptors damage, we hypothesized that IRF localization affects the visual prognosis of BRVO.

In the present study, we reviewed vertical OCT images of ME secondary to BRVO and investigated whether IRF involving the central foveola on vertical OCT images at the initial visit affects the visual prognosis of BRVO.

## 2. Materials and Methods

### 2.1. Patients

This retrospective observational study included patients with ME associated with treatment-naive BRVO. The study protocol was approved by the Institutional Review Board of Shinseikai Toyama Hospital (reference number: 220422-1) and adhered to the tenets of the Declaration of Helsinki. Informed consent was obtained from all the participants. We reviewed the medical and ocular histories of patients with ME attributed to BRVO treated with a single loading dose + pro re nata (one + PRN) regimen of intravitreal injections of aflibercept (Eylea^®^, Regeneron Pharmaceuticals, Inc., Tarrytown, NY, USA) between 8 January 2018, and 4 April 2022, at the Department of Ophthalmology of the Shinseikai Toyama Hospital (Toyama, Japan).

The inclusion criteria were as follows: treatment-naivety with ME within 6 months following BRVO onset and history of treatment with a one + PRN regimen of 2.0 mg/ 0.05 mL aflibercept over 12 months. Patients who had spectral-domain OCT images obtained at the initial visit were also included. The exclusion criteria were as follows: presence of central retinal vein occlusion, hemicentral retinal vein occlusion, significant media opacity and other retinal disorders, such as diabetic retinopathy. Patients who had undergone ocular surgery, such as cataract surgery or vitrectomy within 6 months and during the current study, undergone treatment including intravitreal injection of drugs, such as ranibizumab (Lucentis^®^, Genentech Inc., South San Francisco, CA, USA), aflibercept and triamcinolone acetonide (TA), or who were treated with a sub-Tenon injection of TA for at least 3 months following the treatment, were also excluded.

### 2.2. Examinations

During the study period, all the patients underwent comprehensive ophthalmic examinations, including BCVA measurement using a Landolt C-chart, slit-lamp biomicroscopy, indirect ophthalmoscopy, fundus camera (California, Optos PLC, Dunfermline, UK and/or TRC-NW8, Topcon, Tokyo, Japan), and OCT (Spectralis OCT, Heidelberg Engineering, Heidelberg, Germany and/or RS-3000, Nidek Co., Ltd., Gamagori, Japan). The area of retinal hemorrhage was manually measured by two observers (H.S. and R.M.) using the built-in software of the ultra-widefield fundus camera, and the area was automatically calculated in mm^2^.

Fluorescein angiography was performed to evaluate the perfusion status in all patients using an ultra-widefield fundus camera. A nonperfusion area smaller than five disc diameters was considered perfused [11]. 

BRVO subtype (i.e., major or macular) was determined using a fundus camera based on the vein occlusion: major BRVO, occlusion of one of the four major branch retinal veins; and macular BRVO, occlusion of the veins in the macular region [12].

### 2.3. Definition and Classification of the Two Groups Using Optical Coherence Tomography

Based on the vertical OCT B-scan image through the fovea at the initial visit, we classified the eyes into two groups: the one-side IRF group included eyes with IRF not involving the central foveola (Figure 1), and the both-sides IRF group included eyes with IRF involving the central foveola (Figure 2). If the vertical OCT B-scan detected subretinal fluid (SRF) involving the central foveola with the IRF not involving the central foveola, the eyes were classified into the one-side IRF group in the present study. Two retinal specialists (H.S. and H.I.) independently classified the eyes included in the two groups and were masked to the BCVA at the initial and 12 months visits and other patient information related to the eyes; a senior observer (M.Z.), who was also blinded to the patient data, made the final decision concerning the classification in case of disagreement.

### 2.4. Assessment of the Foveal Ellipsoid Zone Band at 12 Months after the Initial Treatment 

Based on the vertical OCT B-scan image through the fovea at 12 months after the initial treatment, one observer (H.S.) evaluated the foveal EZ band continuity (Figure 3 and Figure 4) and was masked to the BCVA at the 12 months; a senior observer (M.Z.), who was also blinded to the patient data, made the final decision concerning the ambiguous cases.

### 2.5. Statistical Analysis

A biostatistician (K.M.) performed the statistical analyses using SAS version 9.4 software (SAS Institute, Inc., Cary, NC, USA). Normality was evaluated using the Shapiro–Wilk test. The Wilcoxon signed-rank test was used for the paired analyses. The Mann–Whitney *U* test was used for the unpaired analyses. Differences in the categorical data between the two groups were analyzed using Fisher’s exact test. Multiple regression analysis was performed to identify explanatory variables with a statistically significant contribution to logMAR BCVA 12 months after the initial treatment. In this model, logMAR BCVA at 12 months was set as the objective variable. Groups (one-side or both-sides IRF), patient age, baseline logMAR BCVA, baseline CST and presence of SRF were set as explanatory variables upon confirming the absence of multicollinearity among these explanatory variables. All the values are expressed as the mean ± standard deviation. Statistical significance was set at *p* < 0.05.

## 3. Results

### 3.1. Patient Characteristics

Fifty eyes from 50 patients met the study criteria for the analysis. A summary of the patient characteristics is presented in Table 1. The logMAR BCVA and CST at 12 months following the initial treatment showed significant improvement (*p* < 0.001 and *p* < 0.001, respectively) compared to the baseline values.

### 3.2. Comparison of the two Groups According to Foveal Intraretinal Fluid Localization

A summary of the patient characteristics of the two groups is presented in Table 2. The logMAR BCVAs at baseline and 12 months were significantly better (*p* = 0.028 and *p* < 0.001, respectively) in the one-side IRF group compared to that in the both-sides IRF group. No eyes with decimal BCVA less than 0.5 were detected in the one-side IRF group at 12 months. The baseline CST was significantly lower (*p* < 0.001) in the one-side IRF group than in the both-sides IRF group. However, the CST at 12 months did not differ significantly (*p* = 0.26) between the two groups.

### 3.3. Baseline Visual Prognostic Factors Associated with the Visual Acuity at 12 Months

In this study, we considered five explanatory variables that could contribute to the logMAR BCVA at 12 months following the initial treatment. The results of the multiple regression analyses are shown in Table 3. The multiple regression analyses demonstrated that baseline logMAR BCVA and the both-sides IRF group significantly correlated with the 12-months logMAR BCVA (*p* = 0.04, standardized coefficient = 0.28 and *p* = 0.001, standardized coefficient = 0.455). However, patient age, baseline CST and presence of SRF did not correlate with the 12-months logMAR BCVA (*p* = 0.19, *p* = 0.96 and *p* = 0.59).

## 4. Discussion

We investigated the visual prognostic factors in treatment-naive BRVO using OCT images in the current study. Although the CST at 12 months did not significantly differ between the two groups, the logMAR BCVA at 12 months following the initial treatment was significantly better in the one-side IRF group compared to that in the both-sides IRF group. We demonstrated that baseline logMAR BCVA and IRF localization were significantly associated with the 12-months logMAR BCVA. Moreover, no eyes with decimal BCVA less than 0.5 were detected in the one-side IRF group at 12 months. These findings suggest that IRF localization could be a novel biomarker for the visual prognosis of BRVO, along with baseline logMAR BCVA.

In this study, we selected a one + PRN regimen of intravitreal injections of aflibercept. This protocol was similar to that of previous studies [13,14]. LogMAR BCVA and CST improved significantly at 12 months compared to the baseline levels. The total number of injections required for each patient was 3.9 ± 1.6 in this study. These results were similar to those of previous studies [13,14].

Previous studies have described baseline BCVA as a potential visual prognostic factors for BRVO [1,2,3]. Similarly, the baseline logMAR BCVA was significantly associated (*p* = 0.04, standardized coefficient = 0.28) with the logMAR BCVA at 12 months in this study. Considering our results and those of previous studies [1,2,3], patients with better baseline BCVA may have a good visual prognosis in BRVO. Meanwhile, the patient age was not associated with the logMAR BCVA at 12 months in this study. This could be attributed to the sample size [8].

Furthermore, we demonstrated that IRF localization was significantly associated (*p* = 0.001, standardized coefficient = 0.455) with logMAR BCVA at 12 months. This indicates that the visual prognosis of BRVO is worse in eyes with both-sides IRF than in eyes with one-side IRF. In this study, patients with ME secondary to BRVO were treated with a one + PRN regimen of intravitreal injections of aflibercept. Although the CST at 12 months and the total number of injections did not significantly differ between the two groups, the logMAR BCVA at 12 months was significantly worse in the both-sides IRF group compared to that in the one-side IRF group (Table 2). This could be attributed to the significantly greater disruption of the foveal EZ band in the both-sides IRF group compared to that in the one-side IRF group. The integrity of the photoreceptor layer in the fovea is presumably associated with visual acuity in resolved ME secondary to BRVO [15,16]. In this study, eyes with both-sides IRF had severe ME compared to eyes with one-side IRF. Moreover, SRF was significantly more frequently detected in eyes with both-sides IRF than in eyes with one-side IRF. A thicker CST, which demonstrates the severity of ME, could result in the disorganization of the photoreceptor structure and photoreceptor dysfunction [17,18]. In this study, the baseline CST was significantly higher in eyes with both-sides IRF than in eyes with one-side IRF. Thus, we consider eyes with both-sides IRF to more likely present with impaired integrity of the foveal photoreceptors, resulting in EZ disruption.

Previous studies have demonstrated a poor visual prognosis of eyes with SRF secondary to BRVO [19]. However, the effect of SRF on visual prognosis remains unclear [20,21]. In this study, eyes with both-sides IRF more frequently detected SRF at the initial visit compared to eyes with one-side IRF. However, the presence of SRF at the initial visit was not associated with the logMAR BCVA at 12 months. In addition, the effect of CST on the visual prognosis also remains unclear in BRVO [5,6,7,8]. In this study, baseline CST was significantly higher in the eyes with both-sides IRF than in the eyes with one-side IRF. However, baseline CST was also not associated with the logMAR BCVA at 12 months. Thus, we suggest that the IRF localization, but not baseline CST, is important for visual prognosis of BRVO. Since these results could be influenced by the sample size, individual factors and treatment strategies, prospective studies with uniform treatment strategies employing larger numbers of cases are warranted to elucidate the effect of these factors on the visual prognosis of BRVO.

No eyes with decimal BCVA less than 0.5 were detected in the one-side IRF group at 12 months (Table 2). This result suggests that eyes with one-side IRF have a better visual prognosis in BRVO. Considering the treatment strategy, initiating treatment before the IRF involves the central foveola on vertical OCT images could indicate a better visual prognosis in BRVO. IRF localization could aid in the treatment strategies for BRVO; however, further prospective studies are warranted to confirm the usefulness of this treatment strategy.

This study had several limitations. First, this was a retrospective study with an inherent sampling bias. Second, twenty-two eyes included in this study underwent scatter laser photocoagulation for the nonperfusion areas outside the vascular arcades, which may have resulted in a bias in the data. Third, the patients included in this study were followed up for only 12 months. Cases of EZ band recovery on the OCT images and associated BCVA improvement can be detected during follow-up periods over 12 months [22]. Although our study revealed that baseline IRF localization could contribute to the 12-months logMAR BCVA, longer follow-up studies are warranted to confirm our results.

Despite these limitations, our study has several strengths. We can easily classify the presence of one-side or both-sides IRF on the vertical OCT images in acute BRVO. Moreover, IRF localization may aid in the treatment of patients with BRVO.

## 5. Conclusions

Baseline IRF localization on vertical OCT images can aid in the prediction of the visual prognosis of BRVO. Eyes with one-side IRF may have a better visual prognosis in patients with BRVO.

## Figures and Tables

**Figure 1 jcm-11-03540-f001:**
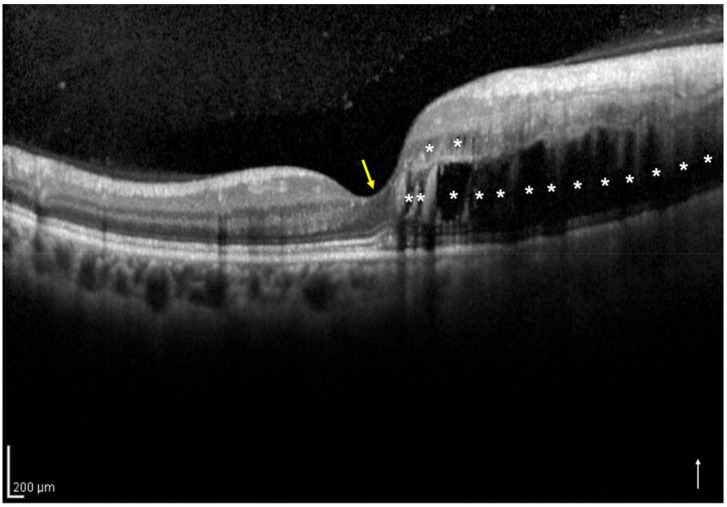
A vertical optical coherence tomography (OCT) image of the right eye with intraretinal fluid (IRF) not involving the central foveola secondary to branch retinal vein occlusion of a 54-year-old woman. The OCT shows IRF (asterisks) not involving the central foveola (yellow arrow). The white arrow indicates the OCT scan direction. This eye was classified into the one-side IRF group.

**Figure 2 jcm-11-03540-f002:**
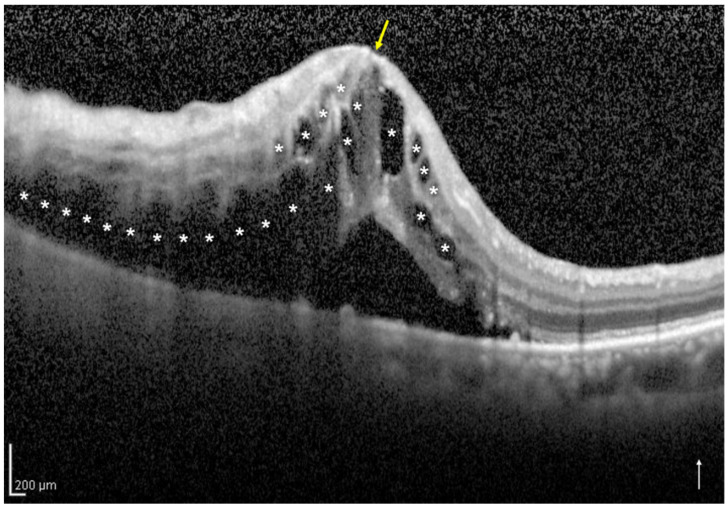
A vertical optical coherence tomography (OCT) image of the right eye with intraretinal fluid (IRF) involving the central foveola secondary to branch retinal vein occlusion of an 85-year-old woman. The OCT image shows IRF (asterisks) involving the central foveola (yellow arrow) and subretinal fluid. The white arrow indicates the OCT scan direction. This eye was classified into the both-sides IRF group.

**Figure 3 jcm-11-03540-f003:**
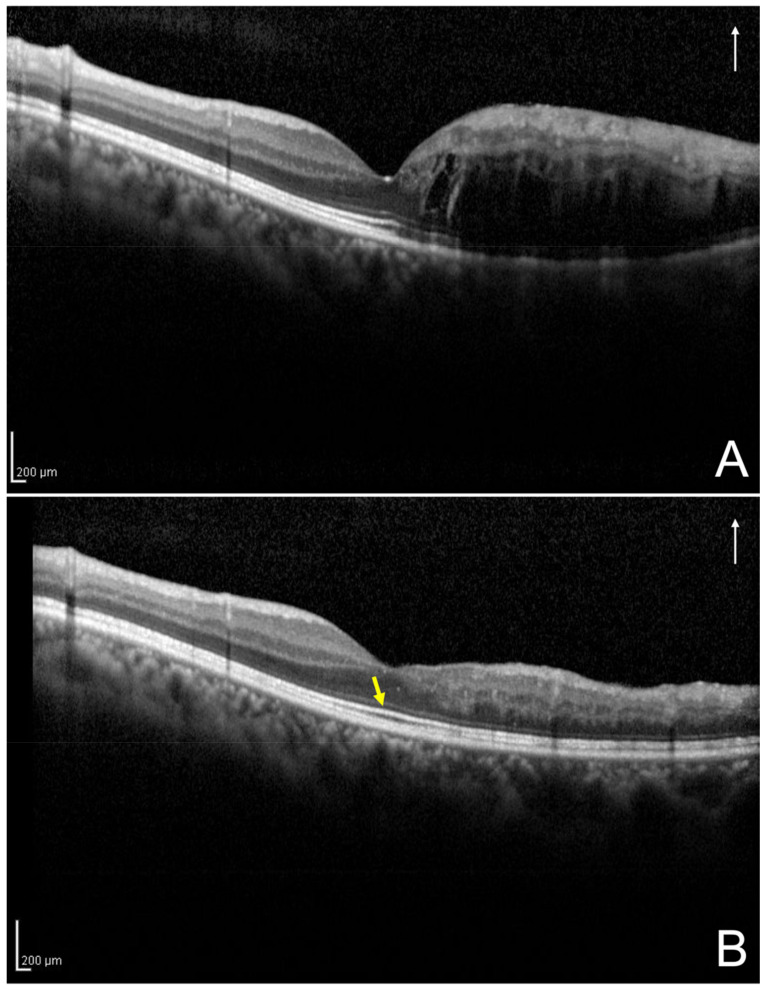
A vertical optical coherence tomography (OCT) image of the left eye secondary to branch retinal vein occlusion of a 43-year-old man at initial visit and 12 months after the initial treatment. (**A**) Based on the initial vertical OCT image, this eye was classified into the one-side intraretinal fluid group. (**B**) The OCT image at the 12 months following the initial treatment shows the absence of disruption of foveal ellipsoid zone band (yellow arrow); the best-corrected visual acuity of the patient was 1.5. The white arrows indicate the OCT scan direction (**A**,**B**).

**Figure 4 jcm-11-03540-f004:**
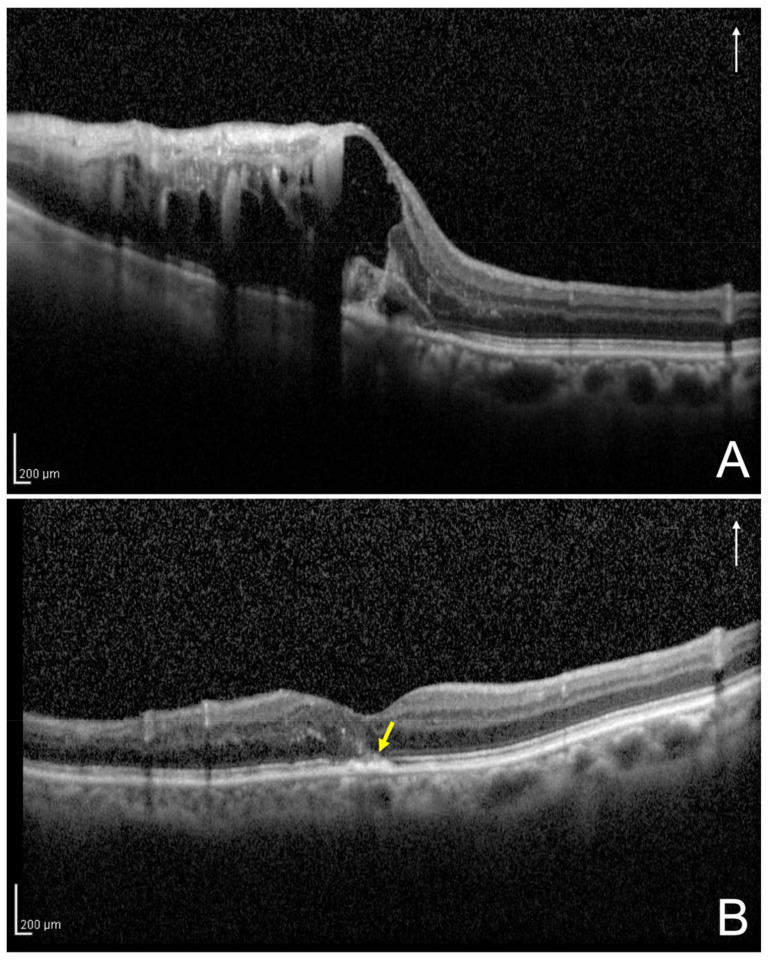
A vertical optical coherence tomography (OCT) image of the left eye secondary to branch retinal vein occlusion of an 87-year-old woman at initial visit and 12 months following the initial treatment. (**A**) Based on the initial vertical OCT image, this eye was classified into the both-sides intraretinal fluid group. (**B**) The OCT image at the 12 months following the initial treatment shows disruption of the foveal external limiting membrane and ellipsoid zone band (yellow arrow); the best-corrected visual acuity of the patient was 0.4. The white arrows indicate the OCT scan direction (**A**,**B**).

**Table 1 jcm-11-03540-t001:** A summary of the patient characteristics and optical coherence tomographic findings.

Parameters	Value	*p*-Value
No. of eyes	50	
Baseline		
Age (years)	67.2 ± 12	
Sex (male/female)	16/34	
Eye (right/left)	22/28	
No. of cases of hypertension (%)	24 (48)	
No. of cases of diabetes mellitus (%)	9 (18)	
No. of cases of dyslipidemia (%)	10 (20)	
BRVO subtype (major/macular)	42/8	
Perfusion status (ischemic/non-ischemic)	32/18	
Hemorrhage area (mm^2^) ^1^	155.3 ± 92.7	
No. of cases of scatter laser photocoagulations (%)	22 (44)	
Duration before initial treatment (weeks)	5.4 ± 5.1	
Total no. of injections	3.9 ± 1.6	
No. of eyes with one-side IRF/both-sides IRF	33/17	
LogMAR BCVA	0.29 ± 0.27	
Central subfield thickness (μm)	512.2 ± 137	
Presence of subretinal fluid (%)	21 (42)	
12 months following the initial treatment		
LogMAR BCVA	0.061 ± 0.2	<0.001 *
Central subfield thickness (μm)	336.5 ± 112.2	<0.001 *
Presence of subretinal fluid (%)	1 (2)	
No. of eyes with foveal EZ band disruption (%)	9 (18)	

* Comparisons between the parameters at the baseline and 12 months were analyzed using the Wilcoxon signed-rank test. ^1^ Two eyes were excluded due to missing data. Data are expressed as the mean ± standard deviation. BCVA, best-corrected visual acuity; BRVO, branch retinal vein occlusion; EZ, ellipsoid zone; IRF, intraretinal fluid; LogMAR, logarithm of the minimum angle of resolution; No., number.

**Table 2 jcm-11-03540-t002:** A summary of the patient characteristics of the two groups.

Parameters	One-Side IRF	Both-Sides IRF	*p*-Value
No. of eyes	33	17	
Baseline			
Age (years)	66.7 ± 11.3	68.1 ± 13.5	0.71
BRVO subtype (major/macular)	26/7	16/1	0.24
Perfusion status (ischemic/non-ischemic)	21/12	11/6	1
Hemorrhage area (mm^2^) ^1^	145.4 ± 94.0	159.4 ± 95.8	0.52
Duration before initial treatment (weeks)	5.1 ± 5.7	6.1 ± 3.7	0.15
Total no. of injections	3.8 ± 1.6	4.1 ± 1.5	0.4
LogMAR BCVA	0.21 ± 0.2	0.43 ± 0.4	0.028
Central subfield thickness (μm)	464.5 ± 125.9	604.7 ± 109.6	<0.001
Presence of subretinal fluid (%)	9 (27.3)	12 (70.6)	0.006
12 months following the initial treatment			
LogMAR BCVA	−0.023 ± 0.1	0.22 ± 0.2	<0.001
Central subfield thickness (μm)	321.5 ± 95.8	365.5 ± 137.4	0.26
Presence of subretinal fluid (%)	0 (0)	1 (5.9)	0.34
No. of eyes with foveal EZ band disruption (%)	1 (3.0)	8 (47.1)	<0.001
No. of eyes with decimal BCVA of ≧0.5 (%)	33 (100)	12 (70.6)	0.003

^1^ Two eyes with one-side IRF were excluded due to missing data. Data are expressed as the mean ± standard deviation. BCVA, best-corrected visual acuity; BRVO, branch retinal vein occlusion; EZ, ellipsoid zone; IRF, intraretinal fluid; LogMAR, logarithm of the minimum angle of resolution; No., number.

**Table 3 jcm-11-03540-t003:** Baseline visual prognostic factors affecting the 12-months visual acuity in the multiple regression analysis.

Explanatory Variables	Coefficient (95% CI)	*p*-Value	Standardized Coefficient
Both-sides IRF (compared to one-side IRF)	0.19 (0.078 to 0.29)	0.001	0.455
LogMAR BCVA	0.2 (0.01 to 0.4)	0.04	0.28
Age (years)	0.003 (−0.001 to 0.007)	0.19	0.166
Central subfield thickness (μm)	0.00001 (−0.0004 to 0.0004)	0.96	−0.008
Subretinal fluid (presence)	0.03 (−0.082 to 0.14)	0.59	0.077

BCVA, best-corrected visual acuity; CI, confidence interval; IRF, intraretinal fluid; LogMAR, logarithm of the minimum angle of resolution.

## Data Availability

The datasets used in this study are available from the corresponding author upon reasonable request.

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
