# Peer review of "Foveal Intraretinal Fluid Localization Affects the Visual Prognosis of Branch Retinal Vein Occlusion"

_jcm, 2022, doi:10.3390/jcm11123540_

Round 1
Reviewer 1 Report
This paper is a study showing that baseline foveal IRF localization on vertical OCT images can be a biomarker for the visual prognosis of BRVO. This is an interesting topic. This is very well structured and very well written. Main objective is clear and well specified.
Is there a relationship between the area of haemorrhage and the form of CME? Some major BRVO have a large or narrow haemorrhage area. Is there more the bilateral-IRF group when the area of haemorrhage is larger? Please indicate the relationship between the area of haemorrhage and the type of CME group.
Reviewer 2 Report
The study is to determine if intraretinal fluid (IRF) exceeding the central foveola was a prognostic factor. However, I find this appellation unclear and it would be more suitable to speak of IRF involving the central foveola.
The patients were divided into two groups in Unilateral and bilateral IRF. I find this denomination confusing as it could indicate both eyes. I suggest authors to choose another terminology.
Line 150: the difference of CST between the two groups is logical as in the group one the foveola is not involved. And the absence of difference at 12 months indicates that there is no difference between the two groups in term of treatment response. The difference in VA seems to be associated to other factors than IRF involving the foveola such as EZ band disruption.
It seems curious that CST was not Prognostic Factors associated with the Visual Acuity at 12 months. EZ band disturbance could also be an important factor of influence.
On other hand, the treatment regimen souhld be discussed and compared to other studies.
Round 2
Reviewer 2 Report
Thanks to authors for their corrections and their efforts.